# OpenReview forum: "Strata-Sword: A Hierarchical Safety Evaluation towards LLMs based on Reasoning Complexity of Jailbreak Instructions"
_ICLR.cc/2026/Conference — Submitted to ICLR 2026_

### Official Review · Reviewer_3WVB · 2025-10-23

**Soundness:** 2
**Presentation:** 2
**Contribution:** 2
**Rating:** 2
**Confidence:** 4

**Summary:**

This paper presents a hierarchical benchmark which is designed to evaluate LLMs under jailbreak attacks of varying reasoning complexity. It introduces reasoning complexity as a new safety dimension composed of logical depth, linguistic ambiguity, and task overhead . Fifteen types of jailbreak attacks are categorized into three hierarchical levels. Experimental results demonstrate that model vulnerability increases with reasoning complexity, revealing clear scaling patterns and cross-lingual asymmetries.

**Strengths:**

1. The introduction of reasoning complexity as a safety dimension is novel and meaningful.
2. The writing is easy to follow.

**Weaknesses:**

1. The definition of reasoning complexity is heuristic. It lacks theoretical derivation to be convinced.
2. The description of the construction, categorization, and other essential details of the 700 attack prompts lacks clarity.
3. The evaluation relies solely on one classifier, QwQ-32B without justification or comparative testing. Moreover, the model’s performance on relevant tasks is not presented in experiments, which may result in biased or unreliable conclusions.
4. The use of a single metric, ASR, is overly simplistic and limits the depth of performance analysis.
5. The multilingual analysis includes only Chinese besides English, making the claimed cross-lingual contribution narrow in scope.

**Questions:**

1. How to ensure the scientific validity of the reasoning complexity hierarchy and its level definitions?
2. What are the key design factors considered when constructing and categorizing the 700 attack prompts?
3. Why is QwQ-32B chosen as the evaluator, and have comparative tests been conducted with other evaluators?
4. Is the classification and annotation process conducted manually or automatically? If manually, how is consistency and fairness ensured among annotators? If automatically, how is model-based annotation accuracy verified?

---

### Official Review · Reviewer_YtyB · 2025-10-29

**Soundness:** 1
**Presentation:** 3
**Contribution:** 2
**Rating:** 2
**Confidence:** 4

**Summary:**

This paper highlights that the general reasoning capabilities of large reasoning models (LRMs) are insufficient to ensure safe responses to jailbreak instructions, which demand even more robust and nuanced reasoning. To address this challenge, the authors introduce "reasoning complexity" as a quantifiable safety dimension, including factors such as logical depth, linguistic ambiguity, and task overhead. Based on this framework, jailbreak attack methods are classified into three levels: Level 1 (Basic Instruction), Level 2 (Simple Reasoning), and Level 3 (Complex Reasoning). Using this categorization, the paper constructs a hierarchical Chinese-English jailbreak safety benchmark named Strata-Sword, and evaluates current LLMs and LRMs to reveal their varying safety boundaries.

Overall, the paper presents an interesting and potentially useful benchmark for safety evaluation. However, the depth of analysis remains limited, and the research would benefit from a more thorough investigation.

**Strengths:**

1. The motivation for building a jailbreak benchmark organized by "reasoning complexity" makes sense and is clearly presented.

2. The curated Strata-Sword benchmark contributes to evaluating the safety boundaries of existing LLMs and LRMs.

**Weaknesses:**

1. The primary concern is that this paper lacks research depth. The three predefined elements of "reasoning complexity" and the corresponding three-tier safety evaluation appear intuitive but are not supported by any guiding principles or theoretical justification to validate the completeness of the categorization.

2. The experiments are somewhat weak. The authors primarily report attack success rates of various LLMs and LRMs across different model families and sizes, without deeper analysis. The five so-called "insights" presented are more akin to surface-level observations derived directly from the evaluation results, which makes the paper resemble an experimental report rather than a substantive research study. It would be much better to involve a deeper investigation into the underlying causes of model behaviors and a discussion of potential strategies to enhance safety alignment.

3. Regarding the proposed Strata-Sword benchmark, the paper lacks a comprehensive comparison with existing safety evaluation benchmarks. This makes it difficult to assess the relative value or novelty of the benchmark within the broader landscape.

**Questions:**

1. In the evaluation setup, why do the authors choose the QwQ-32B model only to assess the safety?

2. For the "Insight 3", what does the "temporal trends" mean? Where's the evidence?

---

### Official Review · Reviewer_2Yr1 · 2025-11-01

**Soundness:** 3
**Presentation:** 3
**Contribution:** 2
**Rating:** 4
**Confidence:** 4

**Summary:**

This work focuses on evaluating model safety—specifically, a model’s ability to consistently refuse malicious instructions. It introduces the Strata-Sword benchmark, which classifies 15 existing jailbreak techniques into three complexity levels based on three dimensions: Logical Depth, Linguistic Ambiguity, and Task Overhead. The study also designs several jailbreak methods tailored to Chinese, demonstrating strong attack effectiveness. By systematically evaluating models on this benchmark, the paper synthesizes the current state of model safety and highlights avenues for improvement.

**Strengths:**

1. The study makes an important point: jailbreak techniques should be stratified by complexity to help us better understand the trajectory of model safety research.

2. It conducts comprehensive experiments, providing systematic ratings of mainstream open-source and closed-source models across 15 jailbreak methods.

3. It summarizes developments in model safety and offers suggestions for future work.

**Weaknesses:**

Main limitation. The evaluation scheme is built on static, heuristic intuition, and the presented results do not always align with the proposed levels. For example, in Table 2, for Vicuna-7B in English settings, the attack success rates for L1 and L2 are very close; similar issues appear for DS-Distill-Qwen2-7B. In Figure 2, the OPPOSING method (categorized as L2) outperforms some L3 methods. These observations suggest that the proposed framework only coarsely reflects attack complexity and does not always withstand closer scrutiny.

Preferred approach. Ideally, we should adopt a dynamic evaluation: mix all jailbreak prompts together and score each prompt using the proposed metrics to produce a prompt-level ranking. This would directly validate the effectiveness of the complexity framework, rather than relying on coarse, method-level categories. It also addresses a practical need: a robust evaluation system must be able to classify all existing jailbreak methods. Heuristic definitions struggle to scale to that goal, whereas a dynamic scoring approach can.

**Questions:**

Refer to our proposed weakness.

---

### Meta-Review · Area_Chair_gxK6 · 2026-01-04

**Summary:**

The paper proposes a safety benchmark for reasoning models' evaluation. It introduces 15 existing jailbreaks and splits them into three complexity levels: logical depth, linguistic ambiguity, and task overhead. Furthermore, the authors also propose some Chinese jailbreak methods. With these systematic evaluations on the benchmark, the paper synthesizes the current state of model safety and highlights avenues for improvement.

Strengths:
1. The view of splitting methods through complexity is interesting.

2. The new attack is also interesting.

Weaknesses:

1. No clear relationship between the complexity level and jailbreak performance. Maybe because its jailbreak method is static and heuristic.

2. Lacks deep analysis.

3. No dynamic evaluations.

4. No comparison with former benchmarks.

The authors do not respond during the rebuttal. Therefore, I suggest to reject.

**Reviewer Concerns:**

The reviewers have concerns on evaluation settings, benchmarks, deep analysis and etc. The authors don't respond in the rebuttal.

**Reviewer Scores:**

The authors don't respond in the rebuttal. Therefore,i think the score won't change

---

### Decision · Program_Chairs · 2026-01-26

Reject